# Wnt10b protects cardiomyocytes against doxorubicin-induced cell death via MAPK modulation

**Lei Chen[‡], Stefano H. Byer [‡], Rachel Holder , Lingyuan Wu , Kyley Burkey, Zubair Shah ***

Department of Cardiovascular Medicine, University of Kansas School of Medicine, Kansas City, KS, United States of America

‡ LC and SHB are co-first authors.
* zshah2@kumc.edu

**Data Availability Statement:** All relevant data are within the paper and its Supporting Information files.

**Funding:** Funding from The Cray Foundation Endowment; CTSA grant from NCATS awarded to

## Abstract

### Background

Doxorubicin, an anthracycline chemotherapeutic known to incur heart damage, decreases heart function in up to 11% of patients. Recent investigations have implicated the Wnt signaling cascade as a key modulator of cardiac tissue repair after myocardial infarction. Wnt upregulation in murine models resulted in stimulation of angiogenesis and suppression of fibrosis after ischemic insult. However, the molecular mechanisms of Wnt in mitigating doxorubicin-induced cardiac insult require further investigation. Identifying cardioprotective mechanisms of Wnt is imperative to reducing debilitating cardiovascular adverse events in oncologic patients undergoing treatment.

### Methods

Exposing human cardiomyocyte AC16 cells to varying concentrations of Wnt10b and DOX, we observed key metrics of cell viability. To assess the viability and apoptotic rates, we utilized MTT and TUNEL assays. We quantified cell and mitochondrial membrane stability via LDH release and JC-1 staining. To investigate how Wnt10b mitigates doxorubicin-induced apoptosis, we introduced pharmacologic inhibitors of key enzymes involved in apoptosis: FR180204 and SB203580, ERK1/2 and p38 inhibitors. Further, we quantified apoptotic executor enzymes, caspase 3/7, via immunofluorescence.

### Results

AC16 cells exposed solely to doxorubicin were shrunken with distorted morphology. Cardioprotective effects of Wnt10b were demonstrated via a reduction in apoptosis, from 70.1% to 50.1%. LDH release was also reduced between doxorubicin and combination groups from 2.27-fold to 1.56-fold relative to the healthy AC16 control group. Mitochondrial membrane stability was increased from 0.67-fold in the doxorubicin group to 5.73 in co-treated groups relative to control. Apoptotic protein expression was stifled by Wnt10b, with caspase3/7 expression reduced from 2.4- to 1.3-fold, and both a 20% decrease in p38 and 40% increase in ERK1/2 activity.

the University of Kansas for Frontiers: University of Kansas Clinical and Translational Science Institute # 5TL1TR002368.

**Competing interests:** These authors declare no competing interests.

## Conclusion

Our data with the AC16 cell model demonstrates that Wnt10b provides defense mechanisms against doxorubicin-induced cardiotoxicity and apoptosis. Further, we explain a mechanism of this beneficial effect involving the mitochondria through simultaneous suppression of pro-apoptotic p38 and anti-apoptotic ERK1/2 activities.

## Introduction

Doxorubicin (DOX) is an effective chemotherapeutic in the anticancer armamentarium [1]. Used frequently against a broad range of neoplasms, the therapeutic reach of DOX is limited by serious cardiomyopathy, with a 9% incidence in the first year following treatment [2]. Childhood cancer survivors have a 12.5% risk of developing heart failure in the 30 years following DOX treatment [2, 3]. These cardiotoxicities manifest in both the acute and chronic windows. Acute, within three days of administration, presents with chest pain, myopericarditis, dysrhythmias, and left ventricular dysfunction [4]. Chronic issues, which can manifest up to 10 years after administration, lead to heart failure with an incidence of 1.7%. The prognosis of congestive heart failure secondary to doxorubicin cardiotoxicity is poor, with ~50% 1-year mortality [5]. Apart from age, where youth and the elderly are at higher risk and have poor cardiovascular health, the main risk factor for cardiotoxicity is dosage, with rates exceeding 18% and 65% once dose reaches 350 and 550mg/m2, respectively [2].

While cancer affects up to one in three individuals, profound improvements in survival have revealed increased rates of cardiovascular disease resulting from cardiotoxic infliction of cancer treatments. Therefore, elucidating the pathological and potential protective molecular mechanisms in DOX-induced cardiomyopathy will provide the groundwork for beneficial therapeutic intervention against the cardiovascular sequelae of chemotherapy. The difficulty is discerning between intended anticancer, DNA intercalation and damage, and cardiotoxic adverse mechanisms. Several hypotheses have been explored. A few of note: DOX induces toxic mitochondrial iron accumulation, reactive oxygen species (ROS) formation via reduction by NADPH dehydrogenase, and DOX inactivation of pro-survival signaling proteins such as extracellular signal-regulated kinase (ERK) [2, 3, 6].

Recent investigations into molecular mitigation of ischemic cardiotoxicity observed that Wnt10b is strongly and transiently induced after myocardial infarction (MI) during the granulation tissue formation phase of cardiac repair, where neovascularization and fibrosis occur [7]. The study found that Wnt10b overexpression in murine cardiomyocytes stimulates further blood vessel growth and suppression of fibrosis formation. Wnt10b belongs to the Wnt family of highly conserved molecules whose role in cardiovascular diseases has not been thoroughly investigated [8, 9].

We investigated whether Wnt10b provides a cardioprotective role in the AC16 human cardiomyocytes by translating these findings from ischemia-induced to DOX cardiomyopathy. Our results demonstrate that Wnt10b protects DOX-induced cell damage, reduces plasma membrane and mitochondrial membrane damage, and mitigates apoptosis by modulation of MAPK signaling of ERK1/2 and p38.

## Methods

### Reagents

Wnt10b recombinant protein was purchased from R&D Systems (Minneapolis, MN). DOX and FR180204 (ERK1/2 inhibitor) and SB203580 (p38 MAPK inhibitor) were obtained

from Sigma (Saint Louis, MO). Antibodies specific to Bcl-XL, Bcl2, p-ERK1/2, ERK1/2, p-p38 MAPK, p38 MAPK, cleaved Caspase 3, and β-Actin were purchased from Cell Signaling Technology (Boston, MA). 3-(4,5-Dimethylthiazol-2-yl)-2,5-Diphenyltetrazolium Bromide (MTT) and JC-1 assay kit were purchased from Invitrogen (NY, USA). LDH-Glo™ Cytotoxicity Assay was purchased from Promega (Madison, WI). A cell death staining kit was obtained from Roche (Nutley, NJ). A cell fixing and staining kit was purchased from BD (San Jose, CA).

## Cell culture

AC16 Human Cardiomyocyte Cell Line was purchased from Sigma (Saint Louis, MO) and maintained in DMEM/F12 with 4.5 g/L glucose supplemented with 10% (v/v) fetal bovine serum (FBS). Cells were cultured to 80% confluence in a 100 mm dish at 37˚C with 5% CO2 before passage and seeding for experiments.

For induction of cell death and apoptosis, AC16 cells were exposed to different concentrations of DOX for various times as indicated, following co-treatment with Wnt10b. The dose of Wnt10b was chosen based on published literature. The 200 nM of 48 hours of DOX-treatment was chosen based on our experiment, as shown in Fig 1A.

## MTT: Determination of cell viability

MTT assay was performed to evaluate cellular viability. AC16 cells were plated in 96-well plates for 24 hours before experiments. After incubation with the treatments mentioned above, cells were continually cultured for 4 hours under the same conditions in fresh media containing MTT (5 mg/mL). Subsequently, the formazan crystals were solubilized in dimethyl sulfoxide (DMSO), and absorbance was measured at a wavelength of 570 nm.

## Quantifying apoptotic rates: TUNEL assay

Terminal deoxynucleotidyl transferase-mediated dUTP nick end labeling (TUNEL) staining was performed using an in-situ apoptosis detection kit (Roche) according to the manufacturer's protocol. After fixing with 4% paraformaldehyde and staining with TUNEL dye, the TUNEL-positive cells were identified by Leica confocal microscope. Results were expressed as the proportion of TUNEL-positive cells to total cells counted.

## Measurement of mitochondrial membrane potential (ψ): JC-1 staining

Tetraethylbenzimidazolylcarbocyanine iodide (JC-1) (Invitrogen) was used to determine the changes in mitochondrial transmembrane potential. JC-1 exhibits potential-dependent accumulation in mitochondria. In normal healthy cells, JC-1 accumulates and forms dimeric J-aggregates in the mitochondria, giving off a bright red fluorescence. However, when the potential is disturbed, the dye cannot access the transmembrane space and remains in the cytoplasm in its monomeric form (green fluorescence).

Grown in 8-well glass chambers, AC16 cells were cotreated with Wnt10b and DOX for 24 hours. The cells were incubated with JC-1 dye in serum-free media for 15 minutes at 37˚C. The media were then removed, and cells were washed three times using PBS. JC-1 fluorescence was measured to assess the emission shift from green (530 nm) to red (590 nm) in polarized mitochondria at 488 nm excitation.

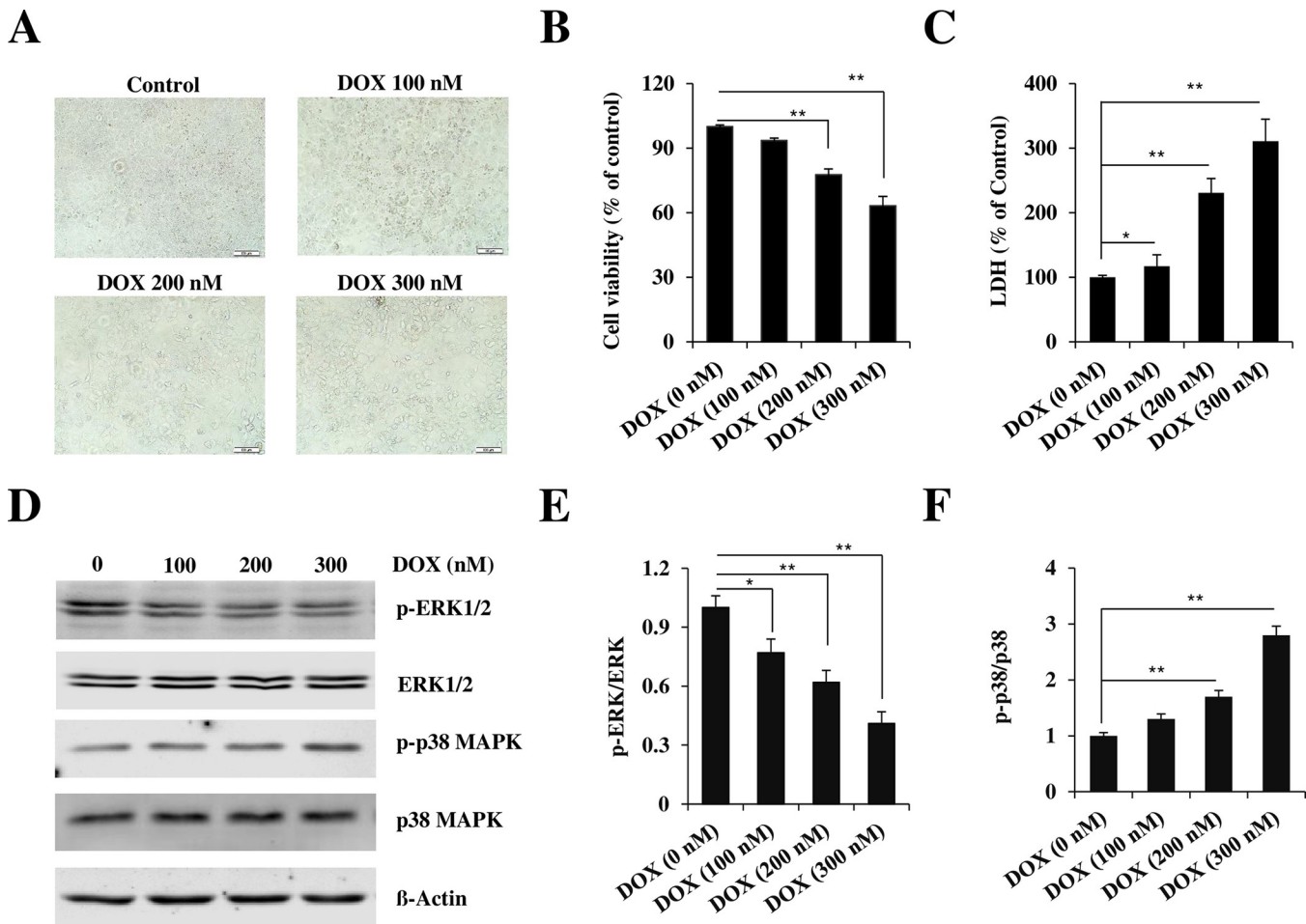

**Fig 1. DOX causes cell toxicity in AC16 cardiomyocyte cells.** (A-B) Results of cell viability analysis of cardiomyocytes that were treated with DOX for 48 hours at the indicated dose. Scale bar = 100 μm. (C) LDH release after DOX treatment at indicated dose for 48 h. (D) Representative immunoblots for protein expression of MAPK kinase and the quantitation of ERK1/2 and P38 activities (E, F). Each data point represents the mean + SD derived from four independent experiments. Differences between groups were considered significant at *p<0.05 and **p<0.01.

### LDH cytotoxicity assay: Assessment of plasma membrane stability

Lactate dehydrogenase (LDH) assays were performed to quantify plasma membrane permeability changes during apoptosis using the LDH cytotoxicity assay kit (Promega). After the proper treatment, the culture media of AC16 cells was used to quantify LDH release.

### Immunoblotting

Whole-cell lysates were prepared in RIPA buffer supplemented with protease cocktail inhibitor mix (Roche). The following quantification by Bradford protein assay and mixing with 5X SDS loading buffer, the lysate was heated to 100˚C for 6 minutes. Samples were subjected to SDS-PAGE electrophoresis using Tris-glycerol buffer. Proteins from the gel were transferred onto the PVDF membrane (Millipore). Membranes were probed with antibodies specific to Bcl-XL, Bcl2, p-ERK1/2, ERK1/2, p-p38 MAPK, p38 MAPK, cleaved Caspase 3, and β-Actin at 4˚C overnight. After primary antibody incubation, membranes were incubated with goat anti-rabbit secondary antibody (LI-COR) and were exposed to the LI-COR image system.

### Cell amplification assay

In total, $1 \times 10^5$ of the AC16 cells were grown in 6-well plates with DMEM/F12 + 10%FBS growth media. After being treated with Wnt10b (200 ng/ml) and DOX (200 nM), the cell number of each well was measured with Invitrogen™ Countess™ II Automated Cell Counter at 12 h, 24 h, 36 h, and 48 h of time points.

### Statistical analysis

Data were collected from experiments of each of the groups and are expressed as means + SEM. Statistical analysis was undertaken using Student's t-test. Differences between groups were considered significant at $^*p < 0.05$ and $^{**}p < 0.01$.

## Results

### DOX induces cellular toxicity in AC16 cells

Changes in cell viability with DOX exposure were qualitatively and quantitatively analyzed via morphological assessment, LDH release assays, and immunoblotting of anti-apoptotic ERK1/2 and pro-apoptotic p38. After 48 hours of DOX exposure, dose-dependent changes were noted in all assays. With worsening pyknosis when comparing control (100 ± 0.77) to 100nM (93.5 ± 1.18), 200nM (77.7 ± 2.67), and 300nM (63.2 ± 4.32) of DOX, as visible in Fig 1A. MTT uptake assays, used to quantify the reduction in cell viability, noted a 6.5%, 22.3%, and 37.2% decrease in 100 nM, 200 nM, and 300 nM of DOX treated group (n = 4, p<0.01) (Fig 1B). Plasma membrane damage increased significantly at 200nM and 300nM DOX, with an increase in LDH release of 231 ± 17.86% and 311 ± 39.03% relative to control (100 ± 2.82), respectively (n = 4, p<0.01) (Fig 1C). Immunoblot analysis revealed that MAPK signaling was affected by DOX in a dose-dependent manner. Whereas DOX concentration increased from control to 200nM and 300nM, the ratio of p-ERK to ERK significantly decreased while the ratio of p-p38 to p38 increased (Fig 1D–1F).

### Cardioprotective effects of Wnt10b in AC16 cells

TUNEL staining and LDH release assays were used to elucidate any cardioprotective properties of Wnt10b in response to DOX toxicity. After 48 hours of exposure to 200 nM DOX, 70.1 ± 3.6% of AC16 cells demonstrated TUNEL-positive results, this reduced to 51.1 ± 4.36% in the cotreated Wnt10b + DOX group (n = 3, p<0.01) (Fig 2A and 2B). LDH assays were used to quantify further the protective effects on the plasma membrane by Wnt10b. AC16 cells co-treated with Wnt10b and DOX (1.56 ± 0.12 folds, n = 5, p<0.01) had a 30% reduction in LDH release relative to cells solely exposed to DOX (2.27 ± 0.1 folds, n = 5,p<0.01) (Fig 2C).

### Wnt10b maintains mitochondrial membrane integrity via restoration of membrane potential (Δψm) and mitigates intrinsic apoptotic mechanisms during DOX exposure

JC-1 staining was used to evaluate if DOX-induced cardiotoxic stress triggers intrinsic cell death through the collapse of the mitochondrial membrane potential (Δψm). Using the ratio of JC-1 staining, with red (healthy) to green (damaged mitochondria) fluorescent cells, to quantify mitochondrial damage, when comparing mitochondrial damage after 200 nM DOX exposure, the Wnt10b pretreated group had a 5.73 ± 1.24-fold increase in the red-to-green ratio (n = 3, p<0.01) (Fig 3A and 3B). Exposing AC16 cells to varying concentrations of Wnt10b and DOX, we assessed the activity of critical modulators of apoptosis: caspase 3, Bcl2,

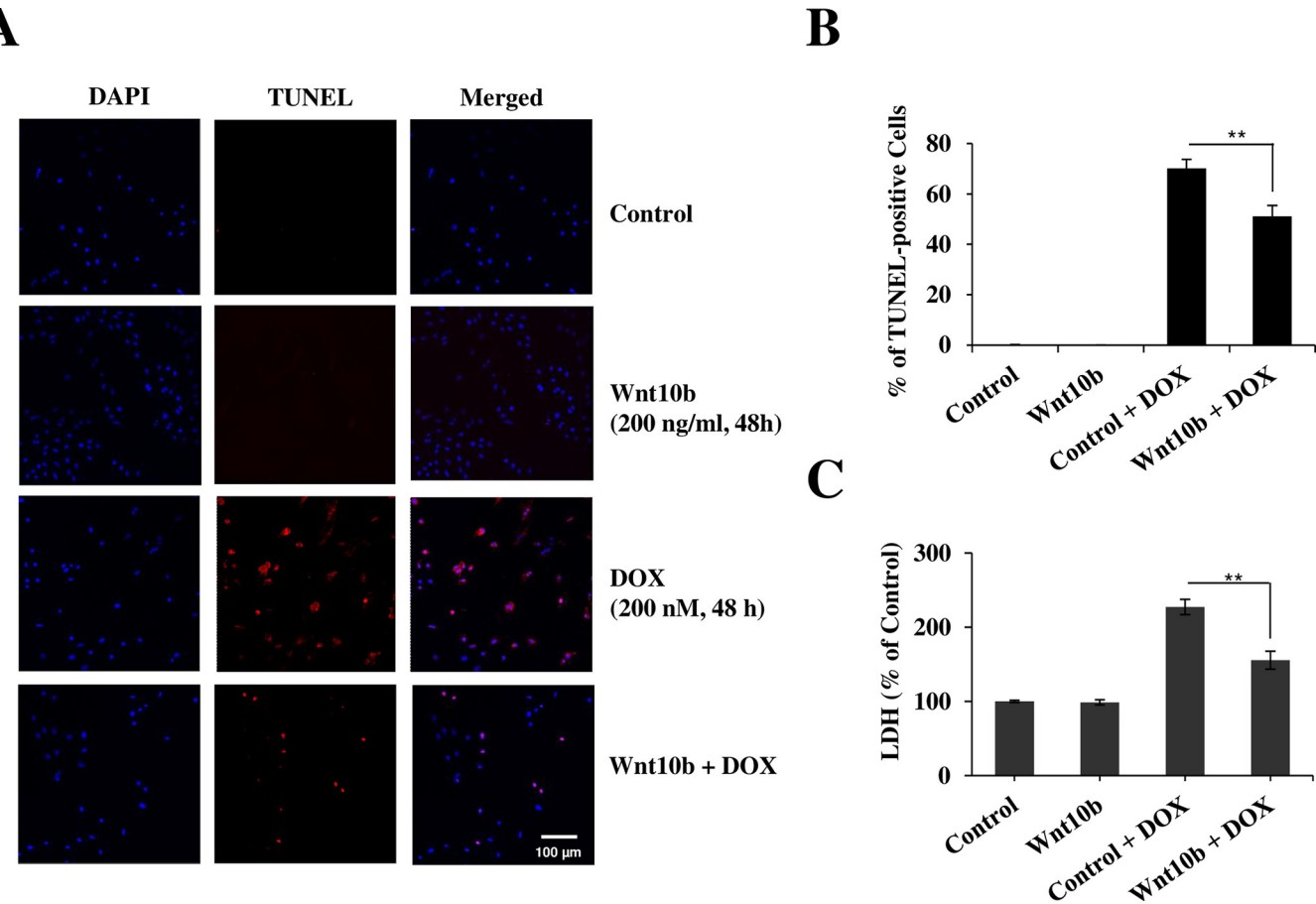

**Fig 2. Wnt10b suppresses DOX-induced apoptosis in AC16 cells.** (A) TUNEL analysis demonstrated cell apoptosis. Scale bar = 100 μm. (B) The bar graph on the right represents the TUNEL positive cells, quantitating the percentage between TUNEL positive cells. Data are mean + SD (n = 3), (C): LDH release quantified after Wnt10b and/or DOX treatment. Data represent as mean + SD (n = 5), **p < 0.01.

and Bcl-xL. After exposure solely to 200 nM of DOX, cleaved caspase three levels increased 2.4 ± 0.11-fold relative to control. However, caspase three activity was attenuated to 1.3 ± 0.05-fold of control after exposure to 50nM of Wnt10b (n = 3, p<0.01) (Fig 3D). The expression Bcl class of anti-apoptotic proteins, while noted to be suppressed by DOX, was significantly increasingly induced by Wnt10b co-treatment.

## MAPK signal finely modulated by Wnt10b as a protective mechanism against DOX-induced cytotoxicity

Immunoblotting was utilized to investigate the interaction of Wnt10b and DOX on the MAPK signaling cascade. As per Fig 4A and 4B, antibody probing revealed that the ratio of phosphorylated-ERK1/2 to ERK1/2 was strongly induced by exposure solely to increasing concentrations of Wnt10b. While DOX suppressed the pERK1/2 ratio, its induction was salvaged with escalating co-treatment of Wnt10b (1.11 ±0.07 folds of 50ng/ml Wnt10b+DOX, and 1.37 ± 0.11 folds of 200ng/ml Wnt10b + DOX respectively vs 0.89+0.12 folds of DOX group). Meanwhile, the opposite trend was found with the ratio of phosphorylated p38 to p38. Exclusive Wnt10b exposure reduced phosphorylated p38 by 12% and 19% in 50 nM (0.88 ± 0.1 folds) and 200 nM (0.81 ±0.09 folds) groups, respectively. DOX strongly induced p38 by a

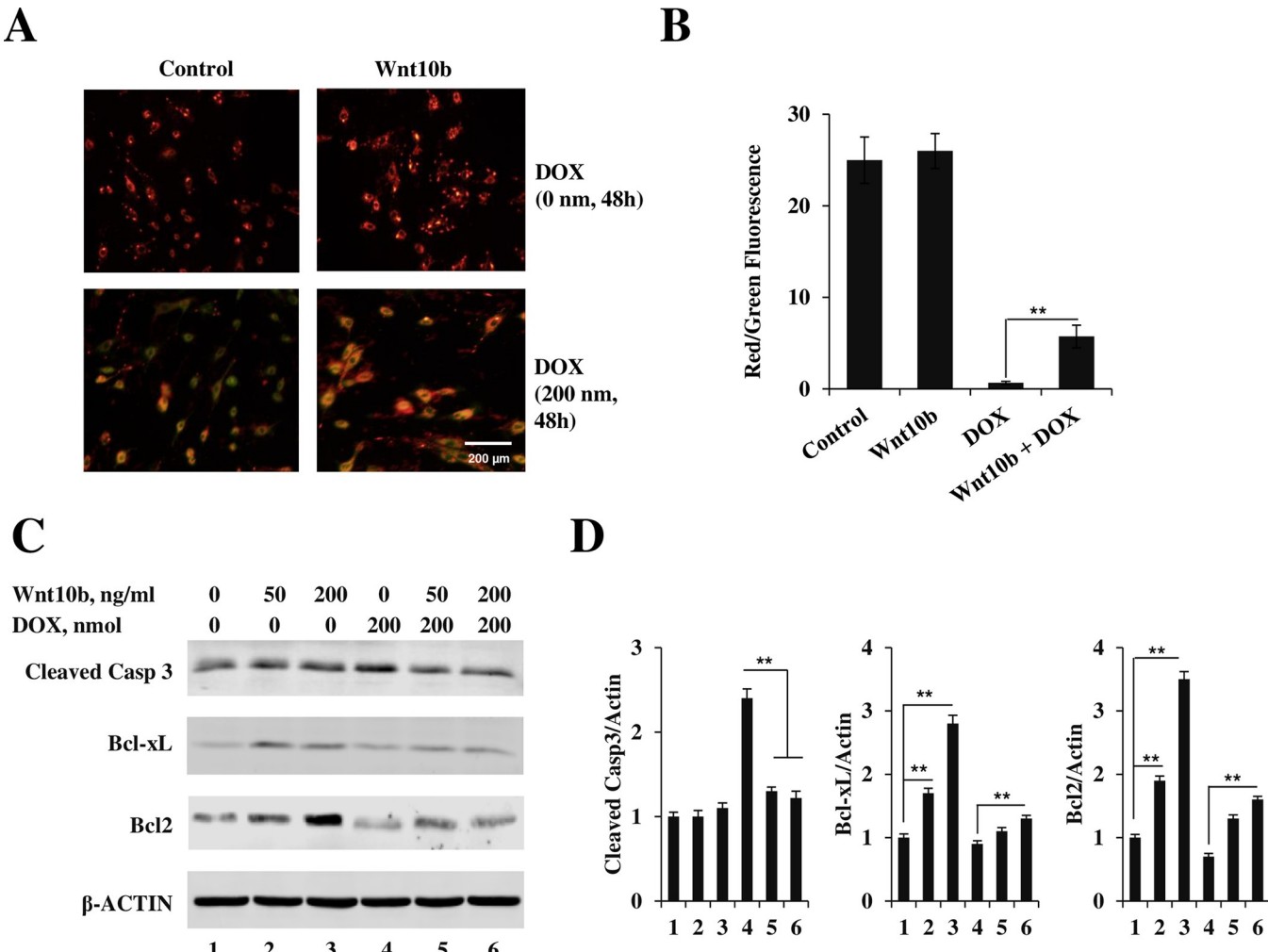

**Fig 3. Wnt10b attenuated DOX-induced damage through mitochondrial modification.** (A) AC16 cells with or without Wnt10b pretreatment were exposed to DOX for 48 hours. JC-1 fluorescence was measured by fluorescence microscopy, (B) the red/green fluorescence ratio represents the mitochondrial membrane potential variation. Scale bar = 200 μm. (C) Immunoblot analysis of Bcl2, Bcl-xL, and cleaved caspase 3 expression levels after Wnt10b and DOX treatment, (D) and the densitometric quantitation of the cleaved caspase 3, Bcl2 and Bcl-xL expression levels compared to the actin control. Data represents mean + SD (n = 3), **$p < 0.01$.

$1.25 \pm 0.1$-fold relative to control ($1 \pm 0.04$), with Wnt10b co-treatment strongly reducing its expression to levels below control.

## Wnt10b protects against DOX-induced cell damage apoptosis via ERK1/2 and p38 MAPK signal regulation

Utilizing inhibitors of the MAPK signaling cascade–ERK1/2 inhibitor FR180204 and a p38 inhibitor SB203580 –in combination with variable concentrations of Wnt10b and DOX treatment with AC16 cells, we investigated the relationship between Wnt10b and MAPK signaling. TUNEL staining and LDH release assays were utilized to quantify the resultant cytotoxic effects of targeted MAPK signaling blockade.

Fig 5A demonstrates that both ERK1/2 inhibition and DOX exposure, both independently and when combined, increase TUNEL-positive cell percentage. With $73.7 \pm 7.65\%$ with sole exposure to 200 nM of DOX, TUNEL-positive cells were reduced to $50.3 \pm 2.82\%$ after co-

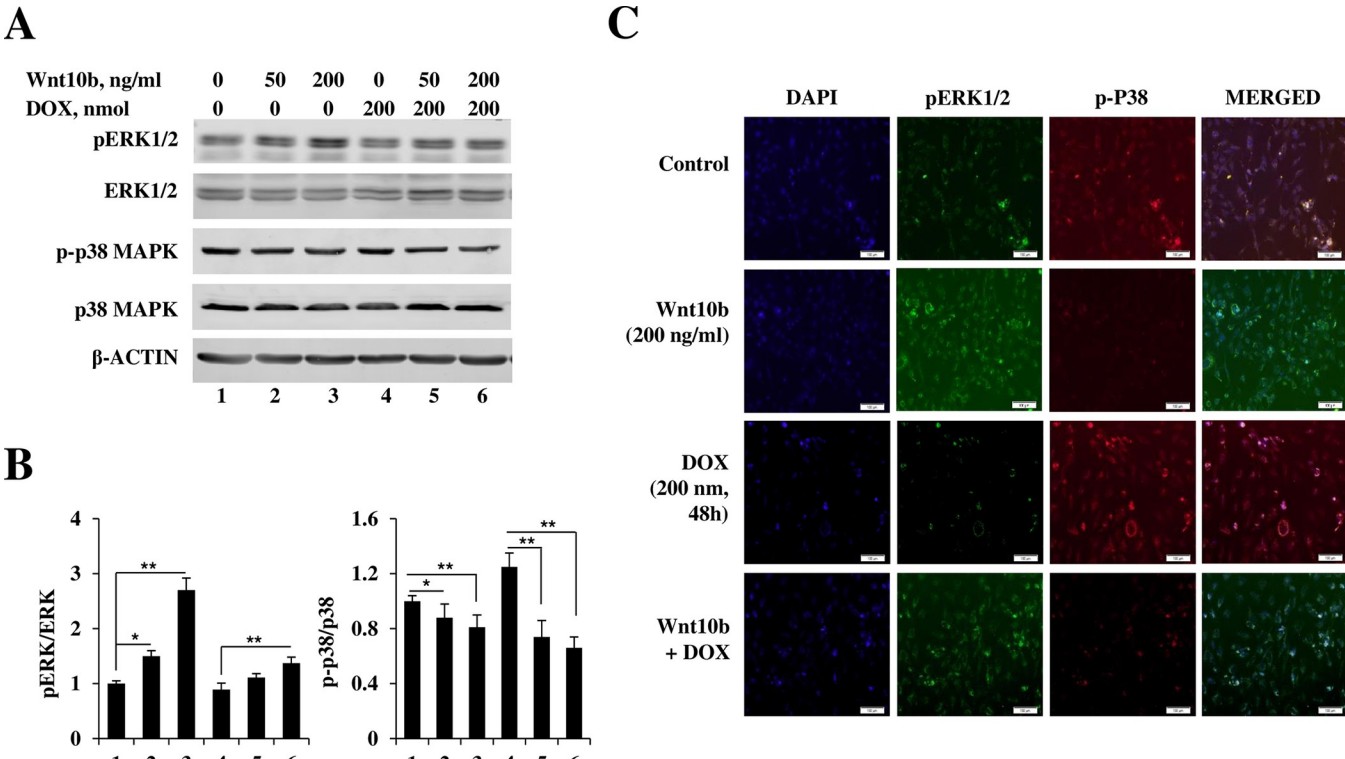

**Fig 4. Wnt10b fine modulation of MAPK signaling in AC16 cells.** (A) Representative immunoblots show pERK1/2 and p-p38 MAPK levels in AC16 cells following Wnt10b and DOX treatment. (B) Densitometric quantitation of pERK1/2 and p-p38 MAPK levels after compared to the expression levels of total ERK1/2 and p38 MAPK. (C) Immunofluorescence staining of p-ERK1/2 AND p-P38 MAPK in AC16 cells after Wnt10b and/or DOX treatment. Scale bar = 100 μm. Data represent mean + SD derived from three independent experiments. *$p<0.05$ and **$p<0.01$.

treatment with Wnt10b. This protective Wnt10b effect was negated by the addition of the ERK1/2 inhibitor FR180204, rebounding the percentage of TUNEL-positive cells to $67.5 \pm 5.37\%$ (n = 4, $p<0.01$) (Fig 5A). These trends were paralleled by LDH release, with ERK1/2 inhibition attenuating the protective effect of Wnt10b (Fig 5A).

Conversely, the addition of p38 inhibitor SB203580 reduced the TUNEL-positive cell percentage relative to the cotreated Wnt10b and DOX AC16 cells from $50.3 \pm 0.58\%$ to $44.5 \pm 2.8\%$ (n = 4, $p<0.01$) (Fig 5B). These results were again paralleled by the LDH release assay, where there was a 18.3% reduction from the Wnt10b cotreated group ($1.64 \pm 0.07$) to the p38 inhibitor addition ($1.34 \pm 0.04$) (n = 5, $p<0.01$) (Fig 5B).

## Discussion

The primary findings of our study suggest 1) Wnt10b preserves cell viability in cardiomyocytes exposed to DOX. 2) Wnt10b engages and enhances MAPK signaling via ERK1/2 upregulation and p38 downregulation, resulting in cell preservation. 3) Wnt10b preserves mitochondrial membrane stability and 4) attenuates intrinsic apoptotic signaling by bolstering pro-survival proteins, Bcl-2 and Bcl-xL.

Doxorubicin (DOX) cardiotoxicity symptoms span a continuum: from subclinical reductions in left ventricular ejection fraction (LVEF) and a rise in biomarkers of cardiac damage to symptomatic congestive heart failure (CHF) and arrhythmias [2]. The cardiovascular toxicity due to DOX is dose-dependent, with a reported incidence of CHF as high as 4.7%, 26%, and 48% at DOX doses of 400, 550, and 700mg/m2, respectively [10–12]. Further, with cardiac

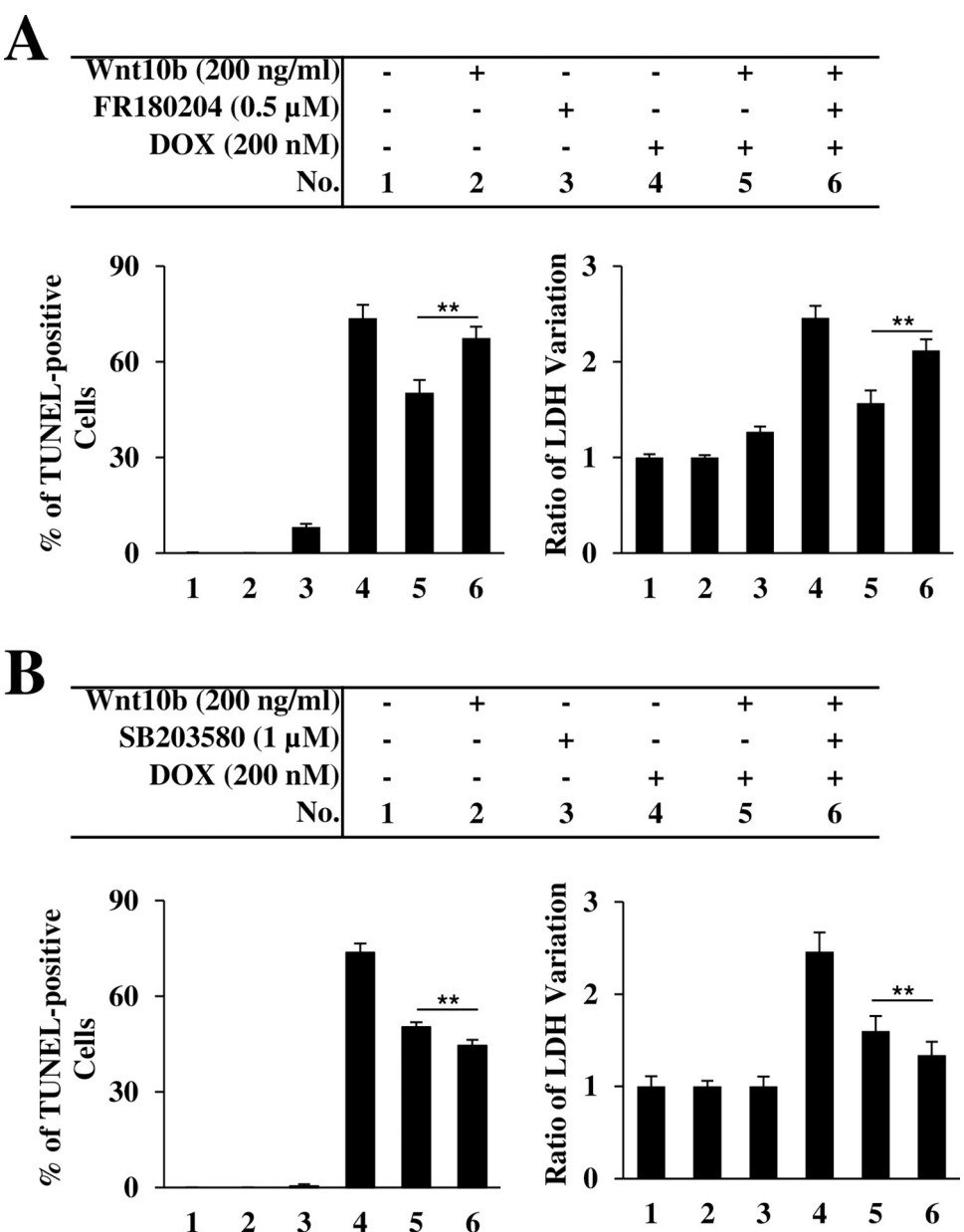

**Fig 5. Wnt10b protects against DOX-induced cytotoxic of AC16 cells via ERK1/2 and p38 MAPK signal regulation.** (A) Quantitation of the apoptosis evaluated by the TUNEL assay and measurement of LDH release after Wnt10b, DOX, and ERK1/2 inhibitor FR180204. (B) Quantify the apoptosis evaluated by the TUNEL assay and measurement of LDH release after Wnt10b, DOX, and p38 MAPK inhibitor SB203580. Each data point represents the mean + SD derived from four independent experiments. **p<0.01.

biomarker rise occurring in up to 40% of patients after administration, the cardiotoxicity of DOX appears to steer cardiomyocyte fate toward apoptosis [13–15]. While the causative mechanisms of cardiac injury have yet to be identified, various hypotheses have been proposed. Our results of human cardiomyocyte cell line exposed to increasing DOX dosage demonstrated visible deformation, plasma membrane rupture per increase in LDH released, and apoptotic cell death.

We investigated the interplay between DOX and MAPK signaling and the potential role Wnt10b had in rescuing cardiomyocytes under DOX-induced duress. The question of

Wnt10b, a ligand triggering the canonical Wnt signaling pathway, as an agent against DOX-cardiotoxicity was inspired by a demonstration of protective effects from ischemic insult. Hatzopoulos et al. found that Wnt10b, released from intercalated discs after myocardial infarction, promotes neoangiogenesis, reduces scar size, and preserves ventricular function. By cotreating human cardiomyocytes with DOX and Wnt10b, we found cell viability was preserved, with significant reductions in apoptosis and plasma membrane stability maintained.

Growing evidence of a Wnt-MAPK connection made it an appealing target of DOX cardiotoxicity investigation. The Wnt-MAPK interplay ranges from physiologic functions, such as chondrogenesis, to malignancy, of which note, pro- and anti-apoptotic proclivities are tumor-dependent [8, 16, 17]. Further, anthracyclines demonstrate pathologic alteration of the Neuregulin-1 (NRG-1) cascade, a pathway upstream of MAPK [2]. Proteins ErbB2 and ErbB4, downstream of NRG-1, have demonstrated mitigation of heart failure and, when activated to heterodimerize, trigger MAPK signaling [18]. Cardioprotective ErbB2 and ErbB4 proteins are significantly disrupted by doxorubicin [19, 20]. Therefore, we investigated MAPK signaling proteins ERK1/2 and p38 to quantify molecular dysregulation secondary to doxorubicin. The apoptotic direction MAPK signaling pathways steer cells is cell type-dependent. Typically, p38 is proapoptotic, and ERK1/2 is protective [21]. Our findings demonstrate progressive suppression of ERK1/2 and conversely increased p38 activity with increasing DOX dose. With co-treatment of the Wnt10b ligand, the effects of DOX were reversed, with ERK1/2 levels rebounded and p38 suppressed. To further isolate the resultant effects on these MAPK proteins from the DOX-Wnt10b tug-of-war, we added ERK1/2 inhibitor FR180204 and p38 MAPK inhibitor SB203580 to the cotreated AC16 cells. Selective inhibition of ERK1/2 rendered the cytoprotective effects of Wnt10b null. At the same time, inhibition of p38 resulted in inverted results, with significant improvement in combating apoptosis. These findings suggest Wnt10b provides cardioprotective support to cardiomyocytes under DOX duress via modulation of the MAPK cascade by upregulation of anti-apoptotic ERK1/2 and suppression of pro-apoptotic p38.

Doxorubicin-induced damage at the mitochondrial level has garnered significant interest. One avenue of investigation is the unintended intercalation of DOX with DNA and topoisomerase 2β (Top2β), rather than the target isoform found in proliferating and neoplastic cells, Top2α [22–24]. The therapeutic binding to Top2α in malignant cells leads to apoptosis, in binding to Top2β–an isoform found in quiescent cells–the cardiomyocyte develops mitochondrial dysfunction and succumbs to apoptosis [2, 22, 25, 26]. Top2β knock-out murine models demonstrated protection against DOX-cardiotoxic mitochondrial demise [27]. While the iron-chelator dexrazoxane, the only FDA-approved prevention of DOX-cardiotoxicity, has been shown to inhibit Top2β, investigations by Ichikawa et al. suggest they are independent targets and pathways for cardioprotection [6]. Further, the group demonstrated that dexrazoxane reduces DOX-induced mitochondrial iron accumulation and reverses cardiotoxicity [6].

The bifurcation in a cell's fate of demise or survival often rests in the in Bcl-2 class of pro- and anti-apoptotic proteins. The anti-apoptotic, Bcl-2 and Bcl-xL keep the pro-apoptotic Bax and Bak from perforating the outer mitochondrial membrane via induction of transmembrane pores [28]. Permeabilizing this membrane opens Pandora's box: the uncoupling of oxidative phosphorylation, which yields rampant reactive oxygen species, and the release of cytochrome c, allowing assembly of key cell executor: caspase-3, culminating in apoptosis [28, 29]. Via voltage-dependent staining, we observed that DOX critically disrupts the mitochondrial membrane potential ($\Delta\psi$m). However, the co-treatment with Wnt10b salvaged not only $\Delta\psi$m, but also the anti-apoptotic proteins Bcl-2 and Bcl-xL. Meanwhile, the expression of apoptotic executor protein caspase 3, which DOX markedly induced, was suppressed by Wnt10b co-treatment. These results suggest that Wnt10b modulates mitochondrial regulatory proteins to preserve cell viability.

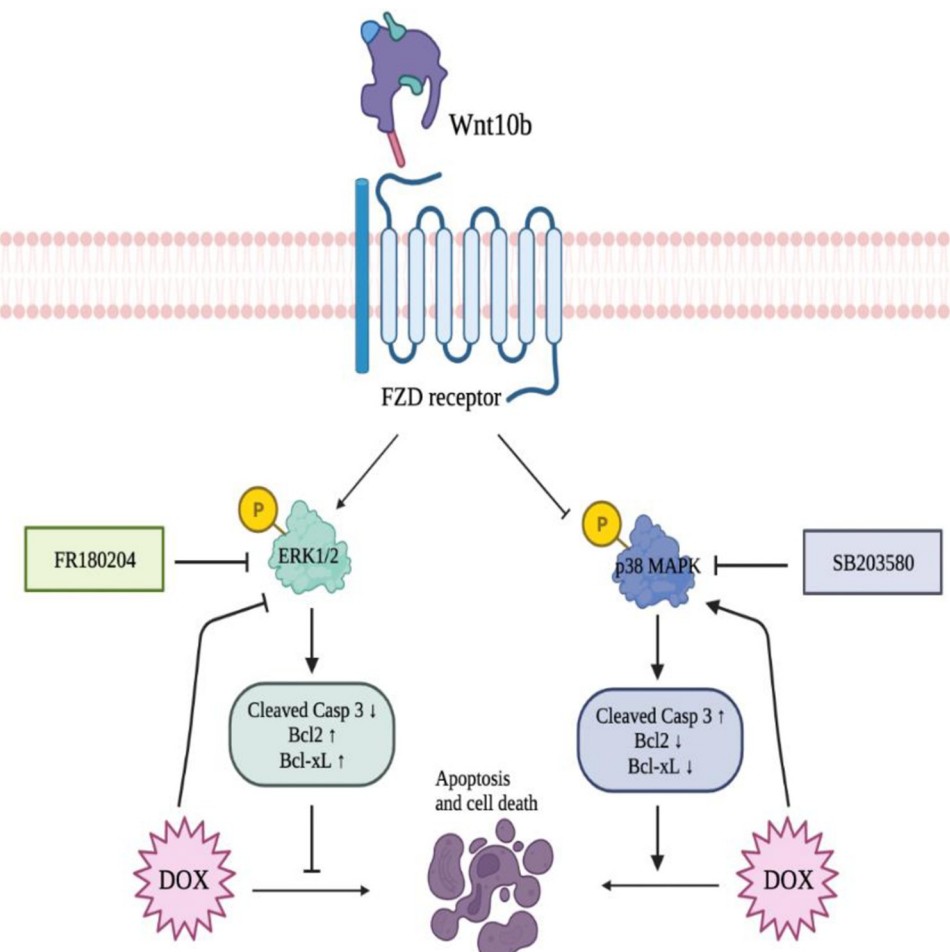

**Fig 6. Proposed signaling cascade and visual summary of findings.** Interplay of doxorubicin (DOX) and Wnt10b on MAPK proteins, ERK1/2 and p38, and resultant downstream effects on apoptotic regulation and cell fate.

We propose Wnt10b as a novel agent to limit DOX-induced cardiotoxicity of AC16 cells by modulation of MAPK signaling via ERK1/2 and p38 pathway and preserving the mitochondrial membrane potential (Fig 6). Given our conclusion being limited by *in vitro* methodology, further studies involving *in vivo* models are required to confirm our findings, elucidate spatiotemporal trends, and explore the molecular mechanism in more detail.

## Supporting information

**S1 Data.**
(XLSX)

**S1 Raw images.**
(PDF)

## Author Contributions

**Conceptualization:** Zubair Shah.

**Data curation:** Rachel Holder, Kyley Burkey.

**Formal analysis:** Lei Chen, Stefano H. Byer, Rachel Holder, Kyley Burkey.

**Funding acquisition:** Stefano H. Byer, Zubair Shah.

**Investigation:** Stefano H. Byer, Rachel Holder, Kyley Burkey, Zubair Shah.

**Methodology:** Lei Chen, Rachel Holder, Lingyuan Wu, Kyley Burkey, Zubair Shah.

**Project administration:** Zubair Shah.

**Software:** Lei Chen.

**Supervision:** Zubair Shah.

**Validation:** Lei Chen.

**Visualization:** Lei Chen, Rachel Holder, Kyley Burkey.

**Writing – original draft:** Stefano H. Byer, Rachel Holder, Zubair Shah.

**Writing – review & editing:** Stefano H. Byer, Zubair Shah.

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
