## [Decision Letter · Decision Letter 0]

8 Jul 2022

PONE-D-22-17005Wnt10b protects cardiomyocytes against doxorubicin-induced cell death via MAPK modulationPLOS ONE

Dear Dr. Byer,

Thank you for submitting your manuscript to PLOS ONE. After careful consideration, we feel that it has merit but does not fully meet PLOS ONE’s publication criteria as it currently stands. Therefore, we invite you to submit a revised version of the manuscript that addresses the points raised during the review process.

Please double check the western blot in figure 1, 3 and 4 as reviewers raised concerns on the western blots.Please check the immunostaining in figures 2, 3, and 4, and address reviewers' concerns. ==============================

We look forward to receiving your revised manuscript.

Kind regards,

Wuqiang Zhu, MD, PhD

Academic Editor

PLOS ONE

6. Please ensure that you refer to Figure 6 in your text as, if accepted, production will need this reference to link the reader to the figure.

Reviewers' comments:

Reviewer's Responses to Questions

**Comments to the Author**

1. Is the manuscript technically sound, and do the data support the conclusions?

Reviewer #1: No

Reviewer #2: Partly

2. Has the statistical analysis been performed appropriately and rigorously? 

Reviewer #1: Yes

Reviewer #2: No

3. Have the authors made all data underlying the findings in their manuscript fully available?

Reviewer #1: No

Reviewer #2: Yes

4. Is the manuscript presented in an intelligible fashion and written in standard English?

Reviewer #1: Yes

Reviewer #2: No

5. Review Comments to the Author

Reviewer #1: I have read with attention and interest the article entitled “Wnt10b protects cardiomyocytes against doxorubicin-induced cell death via MAPK modulation” and I can therefore make the following comments. Wnt signaling pathways play an important role in developmental processes such as tissue patterning, cell differentiation. The authors try to demonstrate that Wnt10b provides defense mechanisms against doxorubicin-induced cardiotoxicity and apoptosis. The paper is well written. However, Inaccurate, and inconsistent data reported. It seems that the data do not support the hypothesis. I believe that the subject of the article is interesting, however I believe that in the form in which the article is presented it is not suitable for publication.

There are several issues the authors need to address to improve the manuscript.

1. In Figure 1D: The shape and slope of p-ERK1/2 band is weird. It seems that the authors rotate the raw band of ERK1/2 by 180-degree, put the rotated band for p-ERK1/2. Could the authors provide the full-sized, uncropped blots as original images?

2. In Figure 2A: There are different background fluorescence between the DOX and Wnt10b+DOX group. To demonstrate Wnt10b suppresses apoptosis in AC16 cells, the authors decrease the fluorescence TUNEL signal, so the red dot signal become very sharp in Wnt10b+DOX group.

3. Figure 3C is not consistent with Figure3D. Figure3C shows that the signal of Bcl-xL band in group 5 and 6 lower than group 4, however Bcl-xL in group 5 and 6 higher than group 4 in figure3D. As you can see the figure3C, the spacing, size, shape, and slope of the bands is different in each blot. This makes it clear that β-Actin likely did not originate from the same blot membrane. The authors simply loaded a different gel and blotted that, so this is not a true loading control. Could the authors provide the full-sized, uncropped blots as original images?

4. In Figure 4C: There are different background fluorescence between Wnt10b, DOX and Wnt10b+DOX group. The authors decrease pERK1/2 green signal in DOX group, then decrease p-P38 red signal in Wnt10b+DOX group, as you can see the red dot signal become very sharp even cannot find any shape of cells. Figure 4A is not consistent with Figure1D. In Figure 4A: When compare the group1(Wnt10b=0; DOX=0) with group4 (Wnt10b=0; DOX=200), the authors will find the ratio of p-ERK to ERK no decreased. β-Actin also likely did not originate from the same blot membrane in Figure 4A.

Reviewer #2: In the manuscript titled “Wnt10b protects cardiomyocytes against doxorubicin-induced cell death via MAPK modulation”, the authors revealed that Wnt10b is a protective factor in the doxorubicin-induced cardiotoxicity and apoptosis AC16 cell model. They explained this mechanism maybe involve p38 and ERK1/2 pathway. In rat cardiomyocytes, doxorubicin administration activated the pro-apoptotic p53, p38 and JNK MAPKs, Bax translocation, disrupted mitochondrial membrane potential, precipitated mitochondrion mediated caspase-dependent apoptotic signaling and reduced viability of cardiomyocytes. Wnt10b in mouse hearts has been shown to improve cardiac tissue repair after myocardial injury, by promoting coronary vessel formation and attenuating pathological fibrosis. There have been many studies on Wnt pathway and doxorubicin-induced cardiotoxicity. I have the following suggestions to improve the manuscript:

• In the background part, the authors mentioned “However, there is a paucity of studies assessing Wnt signaling in doxorubicin-induced cardiac injury.” This statement is not convincing. There are many studies on how to improve doxorubicin-induced cardiotoxicity by modulating this pathway.

• About Figures 1D and Figures 4A, in Figure 1D, the western blot results showed that the expression of p-ERK1/2 is decreased in 200 nM DOX-induced cardiomyocytes. But in Figure 4A, the western blot results showed that the expression of p-ERK1/2 is increased in 200 nM DOX-induced cardiomyocytes. The authors would preferably be able to explain these two opposite results.

• Regarding the representative image of Figure 2A, the authors emphasized that “Wnt10b suppresses apoptosis in AC16 cells”, but the TUNEL signal is higher in the group of Wnt10b, compared with the control group. it will be better if the authors choose a more representative one.

• Regarding the representative image of Figure 4C, it will be more comparable if the author should choose the same exposure time and similar background adjustment. Moreover, if there is an intuitive bar graph it will be better. From the results of Figure 4C, the significant difference is more exaggerated than the results of western blot.

• In the discussion part, the authors can critically discuss the limitations and shortcomings of the previous research and their study and put forward their own prospects and opinions.

• Regarding the format of the article, if the authors can put "Abstract", "Introduction", "Methods", "Results", "Discussion" and "References" into the appropriate place in the article, not at the top of the page, it will be better.

• Please check the format of bibliography and sentences throughout the manuscript. The author's writing language should be more rigorous and in line with scientific facts.

6. PLOS authors have the option to publish the peer review history of their article (what does this mean?). If published, this will include your full peer review and any attached files.

Reviewer #1: No

Reviewer #2: No

---

## [Author Response · Author response to Decision Letter 0]

7 Oct 2022

RESPONSE TO THE COMMENTS OF REVIEWER #1

We would like to sincerely thank the reviewer for their critique, which has helped us to improve the quality of our manuscript. We would also like to thank the reviewer for their kind remarks regarding the quality of work presented in this manuscript. 

Reviewer #1: I have read with attention and interest the article entitled “Wnt10b protects cardiomyocytes against doxorubicin-induced cell death via MAPK modulation” and I can therefore make the following comments. Wnt signaling pathways play an important role in developmental processes such as tissue patterning, cell differentiation. The authors try to demonstrate that Wnt10b provides defense mechanisms against doxorubicin-induced cardiotoxicity and apoptosis. The paper is well written. However, Inaccurate, and inconsistent data reported. It seems that the data do not support the hypothesis. I believe that the subject of the article is interesting, however I believe that in the form in which the article is presented it is not suitable for publication.

There are several issues the authors need to address to improve the manuscript.

1. In Figure 1D: The shape and slope of p-ERK1/2 band is weird. It seems that the authors rotate the raw band of ERK1/2 by 180-degree, put the rotated band for p-ERK1/2. Could the authors provide the full-sized, uncropped blots as original images?

We appreciate the reviewer’s insight in how the figures could be misconstrued. For clarity we have redone the western blots and provided full-sized, uncropped blots as supplemental.

2. In Figure 2A: There are different background fluorescence between the DOX and Wnt10b+DOX group. To demonstrate Wnt10b suppresses apoptosis in AC16 cells, the authors decrease the fluorescence TUNEL signal, so the red dot signal become very sharp in Wnt10b+DOX group.

Thank you for notifying us of the discrepancy. We have rectified the figures.

3. Figure 3C is not consistent with Figure3D. Figure3C shows that the signal of Bcl-xL band in group 5 and 6 lower than group 4, however Bcl-xL in group 5 and 6 higher than group 4 in figure3D. As you can see the figure3C, the spacing, size, shape, and slope of the bands is different in each blot. This makes it clear that β-Actin likely did not originate from the same blot membrane. The authors simply loaded a different gel and blotted that, so this is not a true loading control. Could the authors provide the full-sized, uncropped blots as original images?

Thank you for pointing this out. The data were generated by three independent experiments and quantified by densitometry. We now changed to more representative blot images and list the original images in the supplement figures. 

4. In Figure 4C: There are different background fluorescence between Wnt10b, DOX and Wnt10b+DOX group. The authors decrease pERK1/2 green signal in DOX group, then decrease p-P38 red signal in Wnt10b+DOX group, as you can see the red dot signal become very sharp even cannot find any shape of cells. Figure 4A is not consistent with Figure1D. In Figure 4A: When compare the group1(Wnt10b=0; DOX=0) with group4 (Wnt10b=0; DOX=200), the authors will find the ratio of p-ERK to ERK no decreased. β-Actin also likely did not originate from the same blot membrane in Figure 4A.

Thank you for bringing this to our attention. We generated the data with independent experiments, quantifying it by calculating average density. We’ve added the original images as supplemental figures. 

Regarding the sharp fluorescent images, we believe it is not due to the background and that there are two possible causes. 

One: the fluorescence is not strong enough given that DOX strongly suppresses p-ERK1/2 and wnt10b reduces p-p38. 

Second: DOX induces morphological changes to the cell, altering the fluorescence.

RESPONSE TO THE COMMENTS OF REVIEWER #2

We would like to sincerely thank the reviewer for their critique, which has helped us to improve the quality of our manuscript. We would also like to thank the reviewer for their kind remarks regarding the quality of work presented in this manuscript. 

Reviewer #2: In the manuscript titled “Wnt10b protects cardiomyocytes against doxorubicin-induced cell death via MAPK modulation”, the authors revealed that Wnt10b is a protective factor in the doxorubicin-induced cardiotoxicity and apoptosis AC16 cell model. They explained this mechanism maybe involve p38 and ERK1/2 pathway. In rat cardiomyocytes, doxorubicin administration activated the pro-apoptotic p53, p38 and JNK MAPKs, Bax translocation, disrupted mitochondrial membrane potential, precipitated mitochondrion mediated caspase-dependent apoptotic signaling and reduced viability of cardiomyocytes. Wnt10b in mouse hearts has been shown to improve cardiac tissue repair after myocardial injury, by promoting coronary vessel formation and attenuating pathological fibrosis. There have been many studies on Wnt pathway and doxorubicin-induced cardiotoxicity. I have the following suggestions to improve the manuscript:

In the background part, the authors mentioned “However, there is a paucity of studies assessing Wnt signaling in doxorubicin-induced cardiac injury.” This statement is not convincing. There are many studies on how to improve doxorubicin-induced cardiotoxicity by modulating this pathway.

Thank you for the suggestion, we can see how ‘paucity of studies’ could be misunderstood. We changed the wording to “However, the molecular mechanisms of Wnt in mitigating doxorubicin-induced cardiac insult requires further investigation.”

About Figures 1D and Figures 4A, in Figure 1D, the western blot results showed that the expression of p-ERK1/2 is decreased in 200 nM DOX-induced cardiomyocytes. But in Figure 4A, the western blot results showed that the expression of p-ERK1/2 is increased in 200 nM DOX-induced cardiomyocytes. The authors would preferably be able to explain these two opposite results.

Thank you for pointing this out. 

First, data were generated using independent experiments and quantified using densitometry. 

Second, ERK is a stress-related kinase, the activity of which is easily perturbed by multitude factors. Therefore, its data can often be less stable relative to others.

Regarding the representative image of Figure 2A, the authors emphasized that “Wnt10b suppresses apoptosis in AC16 cells”, but the TUNEL signal is higher in the group of Wnt10b, compared with the control group. it will be better if the authors choose a more representative one.

Thanks for the suggestion. We have corrected this by adding a more representative image. 

Of note, the Wnt10b group sometimes produces subtle TUNEL signal. This is most likely due to growth induced by Wnt10b resulting in higher consumption of nutrition of growth media and leading to cell apoptosis. 

Ultimately, after investigation and calculation, the difference is not significant.

Regarding the representative image of Figure 4C, it will be more comparable if the author should choose the same exposure time and similar background adjustment. Moreover, if there is an intuitive bar graph it will be better. From the results of Figure 4C, the significant difference is more exaggerated than the results of western blot.

We have corrected this background issue in the pERK1/2 of the Wnt10b group image. 

The DOX-induced cell stress sometimes is more serious if cultured on the glass-bottom chamber for immunofluorescence than cultured on cell-culture dish for western blot at the same treating condition. This might be the reason of variation between western blot and immunofluorescence on fluorescence signal, cell shape, western blot densitometry and significance.

In the discussion part, the authors can critically discuss the limitations and shortcomings of the previous research and their study and put forward their own prospects and opinions.

Thanks for the great suggestion on these. We have added this on the discussion part as pink. 

Regarding the format of the article, if the authors can put "Abstract", "Introduction", "Methods", "Results", "Discussion" and "References" into the appropriate place in the article, not at the top of the page, it will be better.

Thanks for the suggestion. We have corrected the positioning now. 

Please check the format of the bibliography and sentences throughout the manuscript. The author's writing language should be more rigorous and in line with scientific facts.

Thanks for the suggestion. We have modified all the bibliography and sentences carefully.

---

## [Decision Letter · Decision Letter 1]

3 Nov 2022

Wnt10b Protects Cardiomyocytes against Doxorubicin-Induced Cell Death via MAPK Modulation

PONE-D-22-17005R1

Dear Dr. Byer,

We’re pleased to inform you that your manuscript has been judged scientifically suitable for publication and will be formally accepted for publication once it meets all outstanding technical requirements.

Kind regards,

Wuqiang Zhu, MD, PhD

Academic Editor

PLOS ONE

Additional Editor Comments (optional):

Reviewers' comments:

Reviewer's Responses to Questions

**Comments to the Author**

1. If the authors have adequately addressed your comments raised in a previous round of review and you feel that this manuscript is now acceptable for publication, you may indicate that here to bypass the “Comments to the Author” section, enter your conflict of interest statement in the “Confidential to Editor” section, and submit your "Accept" recommendation.

Reviewer #1: All comments have been addressed

Reviewer #2: (No Response)

2. Is the manuscript technically sound, and do the data support the conclusions?

Reviewer #1: Yes

Reviewer #2: Yes

3. Has the statistical analysis been performed appropriately and rigorously? 

Reviewer #1: Yes

Reviewer #2: Yes

4. Have the authors made all data underlying the findings in their manuscript fully available?

Reviewer #1: Yes

Reviewer #2: Yes

5. Is the manuscript presented in an intelligible fashion and written in standard English?

Reviewer #1: Yes

Reviewer #2: Yes

6. Review Comments to the Author

Reviewer #1: The authors have adequately addressed my comments, the manuscript has improved from its previous version.

Reviewer #2: The authors has answered my question completely with necessary explanation. The article has been significantly improved. The author has revised the paper as suggested and I recommend publication.

7. PLOS authors have the option to publish the peer review history of their article (what does this mean?). If published, this will include your full peer review and any attached files.

Reviewer #1: No

Reviewer #2: No

---

## [Editor Report · Acceptance letter]

1 Feb 2023

PONE-D-22-17005R1 

Wnt10b Protects Cardiomyocytes against Doxorubicin-Induced Cell Death via MAPK Modulation 

Dear Dr. Shah:

I'm pleased to inform you that your manuscript has been deemed suitable for publication in PLOS ONE. Congratulations! Your manuscript is now with our production department. 

Kind regards, 

on behalf of

Dr. Wuqiang Zhu 

Academic Editor

PLOS ONE